# Research Progress on Emerging Polysaccharide Materials Applied in Tissue Engineering

**DOI:** 10.3390/polym14163268

**Published:** 2022-08-11

**Authors:** Chunyu Su, Yutong Chen, Shujing Tian, Chunxiu Lu, Qizhuang Lv

**Affiliations:** 1College of Biology & Pharmacy, Yulin Normal University, Yulin 537000, China; 2Guangxi Key Laboratory of Agricultural Resources Chemistry and Biotechnology, Yulin 537000, China

**Keywords:** polysaccharide, composite materials, biological scaffolds, wound repair, tissue engineering

## Abstract

The development and application of polysaccharide materials are popular areas of research. Emerging polysaccharide materials have been widely used in tissue engineering fields such as in skin trauma, bone defects, cartilage repair and arthritis due to their stability, good biocompatibility and reproducibility. This paper reviewed the recent progress of the application of polysaccharide materials in tissue engineering. Firstly, we introduced polysaccharide materials and their derivatives and summarized the physicochemical properties of polysaccharide materials and their application in tissue engineering after modification. Secondly, we introduced the processing methods of polysaccharide materials, including the processing of polysaccharides into amorphous hydrogels, microspheres and membranes. Then, we summarized the application of polysaccharide materials in tissue engineering. Finally, some views on the research and application of polysaccharide materials are presented. The purpose of this review was to summarize the current research progress on polysaccharide materials with special attention paid to the application of polysaccharide materials in tissue engineering.

## 1. Introduction 

Polysaccharides are macromolecules formed by a certain number of monosaccharide residues connected under the interaction of glycoside bonds, which are distributed in plants, animals, and microorganisms and play an extremely important role in organisms and food packaging [1,2]. Polysaccharides are also an essential component of all organisms. Its existence maintains homeostasis inside the living body and is indispensable to the existence of the organism itself. Polysaccharides have low toxicity and stable properties as well as rare medical value. Materials made from polysaccharides have the advantages of stable structure, good biocompatibility, and good biodegradation, and they play an important role in the repair of skin wounds, bone defects, cartilage, and arthritis [3].

Polysaccharide materials play an important role in tissue engineering because of their good biocompatibility, stability, and easy chemical modification. At present, a wide variety of polysaccharide materials have been discovered, including sodium alginate, chitosan, hyaluronic acid, chondroitin sulfate, and carrageenan, which have extensive and important applications in tissue engineering [4,5]. For example, chitosan can effectively repair skin wounds on damaged animals, thus improving the recovery efficiency of relevant cells [6]. Hyaluronic acid (HA) has a significant application in bone tissue regeneration [7]. Chondroitin sulfate (CS) also plays an important role in tissue engineering applications due to its advantages of good biocompatibility and lack of side effects [8]. There are many more examples of the application of emerging polysaccharide materials, which will be mentioned in this paper.

At present, research on polysaccharide materials is still ongoing, and the practical application of newly reported polysaccharide materials in tissue engineering is becoming increasingly abundant [9]. Therefore, the purpose of this review was to state the scientific knowledge of related polysaccharides and polysaccharide materials and to summarize the current research and application content of polysaccharide materials. In particular, we highlighted applications of emerging polysaccharide materials in the field of tissue engineering. Firstly, we introduced polysaccharide materials and their derivatives, and then summarized and detailed the physicochemical properties of polysaccharide materials and their modification and application. Secondly, we discussed the processing methods of polysaccharide materials including the processing of polysaccharide hydrogels, microspheres, and membranes. Furthermore, the application of polysaccharide materials in tissue engineering is summarized. Finally, we present our own views on the research of polysaccharide materials. It is hoped that this review will serve as a reference and help researchers in the field of polysaccharide materials as well as provide an important basis for the research of polysaccharide materials and related fields.

## 2. Polysaccharide Materials and Derivatives

Polysaccharides are a hot topic in current research. They are derived from organisms (including animals, plants, and microorganisms) and can also be synthesized in vitro [10]. The presence of polysaccharides plays an important role in maintaining the homeostasis of the internal environment and in maintaining the normal levels of cells. In addition, polysaccharide materials are cheap and have rare properties such as bacteriostaticity, ease of chemical modification, and good biocompatibility and biodegradability [4]. Although there are significant differences among different kinds of polysaccharide materials, the corresponding structural formulas of polysaccharides are presented in Table 1. However, amorphous hydrogels, microspheres, membranes, fibers, and microneedles processed by modified polysaccharide materials can be effectively applied in biological tissue engineering such as for bone defects; skin trauma; and the cardiovascular, nervous, and reproductive systems, as mentioned in the following article [11]. In addition, the composite materials and biological scaffolds prepared by the interaction of polysaccharides with other materials are also important in the application of tissue engineering [8].

### 2.1. Sodium Alginate

Sodium alginate (SA) is a linear anionic polysaccharide that is easily soluble in water, usually presents in a white or pale yellow in color, is derived from brown algae, such as kelp, and its molecules are composed of β -D-mannuronic (M) and α-L-guluronic (G) polyanionic linear copolymers formed according to (1→4) glycosidic bonds [19]. Due to its innate advantages of nontoxicity, biocompatibility, and biodegradability, SA is favored by researchers in tissue engineering and has been extensively studied and applied in bone, cartilage, and skin [4]. Sodium alginate is non-toxic and can be degraded directly in the human body. For example, Niranjan et al. prepared a poly vinyl alcohol (PVA) /SA/ Titanium dioxide (TiO_2_)-Curcumin (CUR) composite patch based on SA and PVA using a gel-casting method modified with Cur and TiO_2_, which proved to be effective in the treatment of wound healing [20]. Similarly, a good way to process SA into composite scaffolds is to take advantage of SA’s property of ease of chemical modification. For example, Yang et al. prepared a silk fibroin (SF)/ hyaluronic acid (HA) /SA composite scaffold with good stability and excellent mechanical mobility by the interaction of SF, HA, and SA using the freeze-dry method. It imparted a good effect in terms of the repair of wounds [21] (Figure 1A). The structural formula of sodium alginate is shown in Table 1 [12].

### 2.2. Chitosan

Chitosan is a linear polysaccharide formed by the interaction of N-acetyl-D-glucosamine and β-glucosamine, which widely exists in the cell walls of arthropods [24]. It is itself a linear glucosamine, and it is the only alkaline polysaccharide with a positive charge discovered thus far. Chitosan has excellent properties and functions such as biodegradability, biocompatibility, is bacteriostatic, provides ease of chemical modification, and has the ability to effectively promote wound healing [25]. Common chitosan derivatives include N-carboxymethyl chitosan, tyrosine glucan, and hydroxyalkyl chitosan, some of which have an extremely important research status in related fields [26]. Chitosan modified by photopolymerization and a composite of other substances to obtain polysaccharide materials play an important role in the field of tissue engineering [27,28]. For example, Shao et al. used fibroin as the basic material and then cross-linked it with photopolymerized chitosan to obtain a biohydrogel, which could effectively improve the accumulation rate of transforming growth factor-β (TGF-β). This can play an important role in cartilage tissue stem cells [29]. Biological scaffolds combine the excellent properties of various basic materials, and the preparation of polysaccharide composite scaffolds has an important application in the field of tissue engineering. For example, Yu et al. used chitosan amine groups as materials, and after modification with methacrylic anhydride, they first prepared methacrylamide chitosan (MC) and then cross-linked it by performing free radical polymerization to form a scaffold. The MC scaffold showed good biodegradability, could penetrate neurites, provided enhanced neuron viscosity, and promoted neurite renewal. The scaffold has a wide range of applications and excellent effects in the field of tissue engineering and neurology [30] (Figure 2A). Ma et al. freeze-dried chitosan, collagen, and glutaraldehyde (GA) for modification to prepare a composite scaffold. It presented excellent stability, good biocompatibility and the effective promotion of cell proliferation, meaning it can be used for repairing skin trauma [31] (Figure 2B). Gull et al. used the fabrication of novel vinyltrimethoxy silane (VTMS) to modify chitosan and polyvinyl pyrrolidone to prepare composite hydrogels, which were used in tissue engineering such as in skin wound repair [32]. The structural formula of chitosan is shown in Table 1 [3].

### 2.3. Hyaluronic Acid

Hyaluronic acid (HA) is a type of natural linear straight-chain polysaccharide alternately linked with glucuronic acid and N-acetylglucosamine, which widely exists in animal joint fluid, connective tissue, cartilage, and eyeball vitreous, and it is insoluble in organic solvents but soluble in water and has excellent water absorption abilities. It has good performance in buffering, lubrication, and tolerance and has excellent water retention, biocompatibility, and a very wide and effective application in tissue engineering, especially in the treatment of osteoarthritis. For example, Gao et al. combined hyaluronic acid hydrogels with nanocrystals to produce a composite hydrogel polysaccharide composite that can achieve the sustained release of camptothecin for the treatment of inflammatory arthritis [34]. Shin et al. modified HA with methotrexate (MTX) to prepare an HA–MTX composite material, but the therapeutic effect of this material on rheumatoid arthritis (RA) was affected by the pH of the buffer solution, and the drug release effect was much better in a weak acidic condition than in neutral and alkaline conditions [22] (Figure 1B). The structural formula of hyaluronic acid is shown in Table 1 [3].

### 2.4. Chondroitin Sulfate

Chondroitin sulfate (CS) is a glycosaminoglycan widely distributed in the cartilage tissues of pigs, cattle, and chickens. At room temperature, chondroitin sulfate is found in a powder form, is easily soluble in water, and insoluble in organic solvents but is hydrolyzed in acidic solutions to generate hexosamine and glucuronic acid. CS itself has good degradability. CS has analgesic and cartilage regeneration functions, and its related salt compounds have good thermal stability [35]. The sugar chain of CS is formed by the polymerization of disaccharides composed of N-acetylgalactose and D-glucuronic acid [36], and an important research direction of CS is its prevention and treatment of osteoarthritis [37]. For example, Clegg et al. used CS and glucosamine in the experimental treatment of knee joint pain, suggesting that the combination of CS and glucosamine had a better effect on the treatment of knee joint pain than the use of either of the two alone [38]. Bauerova et al. used different doses of CS in experiments and finally concluded that CS had a significant effect on reducing the pain of adjuvant arthritis [39]. Chang et al. used chondroitin sulfate and type II collagen as the basic materials, and then modified them with Poly (ε-caprolactone) (PCL) to prepare a composite polysaccharide material scaffold for cartilage tissue application [33] (Figure 2C). The chemical structure of 6-chondroitin sulfate is shown in Table 1 [40].

### 2.5. Carrageenan 

Carrageenan (CRG) is a type of sulphuric acid polysaccharide separated from red algae and other substances. It is generally brown or white granular. It has the advantages of good biocompatibility, good stability, and excellent anticoagulation, and it is insoluble in cold water and organic solvents but can exist in large quantities in hot water. There are many types of CRG. Common carrageenan can be divided into a μ type and κ type, γ type and λ type, etc. Different types of carrageenan have different configurations, but they all have approximately 22–35% sulfate groups inside their structure, which are widely used as gels, thickeners, and stabilizers in food and the pharmaceutical industry [5]. Similar to chondroitin sulfate, carrageenan is also widely used in tissue engineering [40]. For example, Tavakoli et al. prepared a kappa–carrageenan gel with photo-crosslinking technology. Fourier transform infrared spectroscopy (FTIR) and other technologies confirmed that this composite carrageenan gel can be effectively used for the in situ repair of soft tissue and skin trauma, and it has good prospects for development and application in the future [41]. Khan et al. used the freeze-dry method to combine carrageenan, graphene, acrylic acid, and hydroxy-phospholime to prepare a porous composite scaffold, which can be effectively used for bone tissue simulation and drug evaluation in vitro [23] (Figure 1C). The structural formula of κ-carrageenan is shown in Table 1 [5].

### 2.6. Xanthan Gum

Xanthan gum (XG) is versatile in the tissue engineering field of anionic polysaccharides. It widely exists extracellularly in rapeseed yellow unicellular fungi, its normal temperature condition is found as a white or light-yellow powder, it is easily soluble in water, and its thickening, thermal stability, and salt-tolerant sexual functions are outstanding. In addition, polymer xanthan gum has obvious implications in the construction of chemical networks and can be used in cell scaffold construction and drug carrier applications [14,42]. Through the action of related methods, XG can be compounded with other materials, and the prepared polysaccharide materials can be widely used in the field of tissue engineering [43]. For example, Kumar et al. used the freeze-dry method, with XG and SA as the raw materials, and then modified them with cellulose nanocrystals (CNCs) and halloysite nanotubes (HNTs) to prepare a nanocomposite scaffold, which exhibited excellent porosity, compressive strength, and biocompatibility. It is expected to be used as a scaffold in bone tissue engineering repair [44]. Hua et al. took XG, montmorillonite (MMT), and Poly (acrylamide-co-acrylonitrile) (PAAm) as materials and modified them with Fe3+ to obtain an XG/MMT/PAAm composite hydrogel material. The hydrogel integrates electrical conductivity, mechanical properties, and self-healing properties, and it has excellent biocompatibility and other characteristics that play an important role in the field of tissue engineering [45] (Figure 3A). Kumar et al. prepared a sodium alginate-xanthan (AlgX)-CNCs-HNTS composite scaffold using sodium alginate, XG, CNCs, and HNTs as materials via the freeze-dry method. This scaffold had good cytocompatibility and thermal stability and can be used in the field of bone tissue [44] (Figure 3B). The structural formula of xanthan gum is shown in Table 1 [14].

### 2.7. Cellulose

Cellulose is one of the most widely distributed polysaccharides in nature and its molecular configuration is formed by the b-(1,4) glycosidic bond linked to d-glucose. Cellulose is widely distributed in bacterial cell walls or plants with low water solubility but high water absorption, and it can react and hydrolyze under specific conditions. When completely hydrolyzed, it is converted into glucose [48]. The degradation of cellulose requires the action of various enzymes. Although cellulose cannot be absorbed by the human body, it is a helpful digestive aid in the human intestine. In addition, when the graphene/cellulose composite material is obtained by combining cellulose and graphene, the material has good biocompatibility and potential in areas such as bone defects and nerve tissue regeneration [49]. Biocomposites prepared by cellulose will play an important role in the development of tissue engineering materials in the future [50]. For example, Jiang’s research group used carboxymethyl cellulose (CMC) as raw material to prepare the composite scaffold, n-HA/CS/ CMC, via the freeze-dry method. After soaking in phosphate-buffered solution (PBS) and other experiments, the composite scaffold showed good compressive resistance, nontoxicity, and biohistocompatibility, and it could be used as a good substitute for bone defect materials [51]. Ninan et al. also used freeze-dry techniques to prepare cellulose scaffolds with excellent thermal stability, biodegradability, and biocompatibility using pectin, microfibrillated cellulose (MFC), and CMC as raw materials through interaction, which can be used as tissue engineering materials [46] (Figure 3C). Hickey et al. summarized the use of cellulosic biomaterials for tissue engineering [52]. The structure of cellulose is shown in Table 1 [3].

### 2.8. Agarose

Agarose (AG) is a natural polysaccharide derived from red algae, and its basic structure is a long chain alternately connected by 1, 3-linked β-D-galactose and 1, 4-linked 3, 6-anhydro-α-L-galactose [53]. AG is insoluble at room temperature, but it can be dissolved only when heated to above 90 °C in water, and can form a gel after cooling. Because of its nontoxic, biocompatible, adjustable, and thermally reversible gel behavior, AG can play a role in tissue engineering [54]. For example, Ninan et al. used carboxylated agarose (C60) and tannic acid (TA) as substrates. After 10 min in a water bath heated to 70 °C, ZnCl_2_ and NaHCO_3_ were added to modify the material, and after cooling, hydrogels were obtained. The hydrogel is very sensitive to pH and can effectively control and release the tannic acid rate with bactericidal and anti-inflammatory properties under different pH conditions, which can be used for tissue injury repair [47] (Figure 3D). The structure of agarose is shown in Table 1 [15].

### 2.9. Heparin

Heparin is a polysaccharide that exists in animal tissues and is often used as an anticoagulant in clinical operations or diseases such as cardiovascular surgery, cardiopulmonary bypass, and myocardial infarction. Heparin can be degraded by methods such as depolymerization or fractionation, to produce low-molecular weight heparin, which has important applications [55]. Heparin cannot be taken orally and, generally, needs to be injected intravenously. If heparin is processed into stents or used in the development of artificial blood vessels, the developed heparin materials have broad application prospects in tissue engineering [56,57]. For example, Wang et al. modified gelatin with heparin to prepare a gelatin–heparin fiber scaffold, which had good cell activity and application potential in vascular tissues [58]. In a study conducted by Qi et al., heparin effectively inhibited the proliferation of tumor necrosis factor-α (TNF-α), and also indicated the application prospects of heparin in the treatment of RA [59]. Gümüşderelioğlu et al. modified chitosan scaffolds via freeze-dry technology and further improved the scaffolds after the covalent immobilization of heparin. Through actual comparative tests, the scaffolds played a positive role in the proliferation of bone cells and were applied in the field of bone tissue engineering [60] (Figure 4A). The structural formula of heparin is shown in Table 1 [16]. 

### 2.10. Pectin

Pectin is a galacturonic acid polysaccharide that exists in plant cell walls. The degradation of pectin requires the interaction between glycoside hydrolases and polysaccharide lyases. It is often used for gels, thickeners, and emulsifiers in the food industry; At the same time, pectin can also help control blood sugar and blood fat, and it has very good solubility and water retention. Pectin can form a weakly acidic viscous colloidal solution in water at a ratio of 1:20 [63,64,65]. Pectin, as an extremely important substance in polysaccharides, has been widely used in tissue engineering [66,67]. For example, Coimbra et al. used pectin and chitosan as the basic materials and prepared a nontoxic PEC composite scaffold with excellent stability and biocompatibility in an acidic solution by applying the freeze-dry method. The excellent cellular adhesion of this scaffold was confirmed in the toxicity evaluation of bone cells, which can be used for the regeneration of bone tissue defects after injury [61] (Figure 4B). Munarin et al. modified pectin with Arg-Gly-Asp (RGD)-loaded microspheres to prepare composites for bone regeneration [17] (Figure 4C). The structural formula of pectin is shown in Table 1 [17].

### 2.11. Gellan Gum

Gellan gum (GG) is a linear polymer polysaccharide composed of the following three monosaccharides: glucuronic acid, rhamnose, and glucose [68]. The properties and applications of gellan gum are somewhat similar to pectin, but its own acid resistance, enzymatic resistance, stability, and excellent gel performance cannot be surpassed by pectin. Gellan gum is insoluble in cold water and can form a gel with excellent firmness after being dissolved in boiling water and cooled down [69]. If gellan gum is processed into polysaccharide material, it can be widely used in the field of tissue engineering [70,71]. For example, Cerqueir et al. prepared a GG–HA composite hydrogel for promoting cell viscosity and improving skin matrix conditions by interacting with gellan gum and hyaluronic acid. This hydrogel has positive effects on blood vessel regeneration, tissue recovery, and skin injury repair, and it is expected to be used in the repair of skin tissue injury in the future [71]. Oliveira et al. prepared an injectable hydrogel by the interaction of methacrylic acid and gellan gum, which showed positive effects in experiments of bone differentiation of human adipose stem cells (hASCs), and it could be used as a polysaccharide material to induce bone defects [72]. Gellan gum can be used to prepare hydrogels with good biocompatibility, but the stability of hydrogels is not optimal. Therefore, Xu et al. used methacrylate to modify gel-cooling glue gel, following which the modified hydrogel was much more stable and flexible and could be used in other fields such as tissue engineering [62] (Figure 4D). The structural formula of gellan gum is shown in Table 1 [18].

## 3. Processing Method of Polysaccharide Material

There are many ways to process polysaccharide materials, for example, chitosan can be processed into various forms including biological scaffolds, hydrogels, and membranes [69], which can be applied to tissue engineering and other fields [9]. Here, we list materials such as amorphous hydrogels, microspheres, membranes, fibers, and microneedles processed from polysaccharides. They have great prospects for tissue engineering applications such as skin trauma, bone defects and the recovery of blood vessels in chronic wounds, as well as the healing of skin tissues and the development of composite materials. The methods of processing polysaccharides into composites are listed in Table 2.

### 3.1. Amorphous Hydrogel

Hydrogel is a polymer with a three-dimensional polymerization network structure that can be synthesized using physical and chemical methods. Due to amorphous hydrogel’s injectable, less-invasive, good biocompatibility, and degradable characteristics, it has rare advantages in the repair of irregular trauma, cartilage tissue, and other tissue trauma treatment and recovery [73,74]. Many types of hydrogels can form hydrogels in situ, and injectable hydrogels prepared by polysaccharides show a positive effect in tissue engineering fields such as wound repair [75,76,77,78]. For example, Hu et al. modified alginate with the divalent chelate of N-carboxymethyl chitosan (CMC) and epidermal growth factor (EGF), and then they prepared an amorphous hydrogel by inducing electrostatic interaction. This hydrogel showed an ability to speed up wound healing and provided a new strategy for amorphous hydrogel in wound healing [79] (Figure 5A). Lavanya et al. studied the application potential of chitosan-based injectable hydrogels in bone tissue engineering in response to pH value and temperature in depth. If the hydrogel is combined with different biotechnologies, it can be used effectively for bone repair [80].

### 3.2. Microsphere 

Microspheres generally refers to tiny spherical or spheroid particles that can dissolve drugs. Their sizes range from nano to micron, and they are important basic materials. According to the different structures of microspheres, we can classify them into core–shell microspheres [83], homogeneous microspheres [84], and inverse protein microspheres [85], which were systematically summarized in a study by Liu et al. [36]. A variety of polysaccharide materials processed into microspheres have great application value in tissue engineering. For example, Lie et al. prepared a magnetic chitosan (MCS) microsphere using Fe_3_O_4_ nanoparticles and materials, such as Zn^2+^, under the action of microfluidic technology. With the addition of Zn^2+^ and vascular endothelial growth factor (VEGF), this microsphere had a positive effect on injuries. Experimental methods also confirmed that this microsphere can be used for the repair of trauma and has irreplaceable application value in biomedicine [81] (Figure 5B). Lie et al. prepared a composite polysaccharide microsphere by combining HA and AG under microfluidic electrospray technology in in vivo experiments in diabetic mice and confirmed that the microspheres can effectively be applied to the vessel recovery of chronic wounds [86].

### 3.3. Membrane

The membrane is an important structure, and its selectivity plays an irreplaceable role in many fields. Most polysaccharides can be processed into membranes, and the prepared biofilm materials are generally nontoxic and biocompatible and occupy a certain position in the field of tissue engineering [87]. For example, Ma’s research group used SA as the base and modified it with poly (ethylene oxide) (PEO); then, SA/PEO nanofiber membrane was prepared. The membrane showed nontoxicity and biocompatibility, which makes it a good material for application in tissue engineering scaffolds [4] (Figure 5C). Duan et al. prepared a poly (lactide-co-glycolide) (PLGA)–chitosan–PVA nanofiber composite film by cross-pinning PLGA, chitosan, and PVA under the support of a high-voltage power supply. The composite film showed good ductility and promoted the regeneration of fiber cells. It can provide new strategies for the repair of skin trauma [88] (Figure 6A).

### 3.4. Fiber

Fiber is a substance composed of filaments, which can be divided into the following two types: natural fibers and chemical fibers. For example, nylon and polyester are two kinds of fiber that are very common in our daily lives. By combining fibers with related natural polymers, the resulting composites could have important applications in tissue engineering and biomedicine [91]. There are many manufacturing methods for composite polysaccharide fibers [92,93]. Polysaccharide materials can be processed to form fibers, which can be applied in tissue fields such as wound repair [94]. For example, Iwasaki et al. prepared an alginate–chitosan composite polysaccharide fiber by combining alginate and chitosan. Compared with alginate polymer, this fiber improved the adhesion ability of chondrocytes to a greater extent, and it could effectively maintain the morphological normalization of chondrocytes. It is also an ideal polysaccharide material for cartilage repair [95]. At the same time, the construction of nanofibers is also a good processing method. We mentioned two kinds of nanofiber materials in Section 3.3. Nanofibers prepared by chitosan and cellulose and other polysaccharides have unique properties and good applications in tissue engineering [96,97]. For example, Xu et al. prepared a nanofiber material with high water absorption and high speed by connecting pullulan, tannic acid, and chitosan. It could effectively promote the metabolism of fibrous cells, accelerate the growth and development of cells, and it was highly conducive to the repair of severe skin trauma [98].

### 3.5. Microneedle

Microneedling is a new technology combining subcutaneous tissue injection and drug delivery. Drugs injected by microneedles can better improve the absorption of drugs. Compared with general needles, microneedles cause less skin trauma and can effectively reduce the risk of skin infections while reducing the risk of trauma. Microneedles occupy a certain position in tissue engineering precisely because of their painlessness, safety, minimal invasiveness, and portable advantages [99,100]. Microneedles are mainly divided into the following five types: solid microneedles, dissolved microneedles, coated microneedles, hollow microneedles, and hydrogel-formed microneedles [82] (Figure 5D). If polysaccharides are processed into microneedles, they can be used for drug delivery. For example, Yu et al. synthesized an Alg–3-Aminophenylboronic acid (APBA) composite microneedle by combining modified alginate with HA to load insulin. Experimental results in diabetic mice showed that the use of this microneedle more effectively reduced glucose levels, and it is expected to be used for the control of blood glucose levels by subcutaneous drug injection [89] (Figure 6B). Hu et al. prepared a chitosan microneedle array (CSMNA) patch using chitosan as the material modified by VEGF, which could control drug release in an intelligent manner and effectively promote tissue regeneration in wound healing. Thus, it plays a positive role in skin wound repair [101].

## 4. Tissue Engineering Application of Polysaccharide Material

The hydrogels, microspheres, microneedles, and other polysaccharide materials processed by polysaccharides have good compatibility, stability, and degradation as well as high efficiency, leading to the extensive application of polysaccharide materials in the field of tissue engineering. Polysaccharide materials are widely used in skin wound repair, bone defect repair, cartilage repair, arthritis lubrication and pain reduction, cardiovascular system disease treatment, recovery after nervous system damage, reproductive system-related symptom repair, contraception, and other fields. Different kinds of processed polysaccharide materials have different properties. Accordingly, they can be applied to different tissue engineering fields and play different positive roles.

### 4.1. Skin Trauma

The skin is the largest organ of the human body and consists of the following three layers: the epidermis, dermis, and inferior cortex [102]. However, external forces often causes mechanical damage such as skin bleeding, tissue rupture and even defects, which represent a type of trauma that is termed skin trauma. Generally, there are three stages of wound repair, namely inflammation, proliferation, and maturation [98]. Completely repairing skin wounds has remained an urgent aim of researchers. There are many common repair methods, among which wound dressing is a good method for wound repair [103,104], and tissue engineering substitutes can be applied to skin wounds [102]. Polysaccharide materials also play an important role in the repair of skin trauma [6,105]. For example, Guan et al. modified hyaluronic acid methacryloyl (HAMA) with gelatin methacryloyl (GelMA) and formed a new, inexpensive, and biocompatible GelMA/HAMA peptide-coupled patch after UV irradiation that can be effectively used for wound healing of skin and other related tissues [106] (Figure 7A).

Related polysaccharides play a unique role in changing the properties of biological scaffolds because the introduction of hyaluronic acid can significantly improve the properties of scaffolds. For example, using CS, polycaprolactone (PCL), and HA as raw materials, Chanda et al. prepared a double-layer CS–PCL–HA scaffold with good biocompatibility, high stability, and excellent antibacterial properties through the action of electrostatic spinning technology. Compared with the CS–PCL scaffold, this scaffold had better hydrophilicity, degradation, and permeability and can be used to repair skin trauma [107] (Figure 7B).

Skin burns generally leave inevitable scars, and effectively eliminating these scars is a target that scientific researchers have pursued. Polysaccharide hydrogels have many attractive characteristics for application in the repair of skin tissue burns. For example, Huang et al. used rigid rod-like dialdehyde-modified cellulose nanocrystals (DACNC) and water-soluble carboxymethyl chitosan (CMC) and prepared a nano-self-healing composite hydrogel that was injected into the wound, where it completely filled the wound. Then, after being dissolved by an amino acid solution, it not only accelerated the healing of the wound but also effectively prevented the formation of scars [108] (Figure 7C).

### 4.2. Bone Defect

Bone defect refers to the loss of bone tissue caused by a variety of factors, and it is a serious problem for organisms. For the treatment of bone defects, autologous bone grafting, allografts, and nonmetal and metal implantations are generally the most common clinical treatment methods, although they still have some shortcomings at present [110,111]. Polysaccharides can be used in the treatment of bone defects. Composite polysaccharide materials prepared by combining polysaccharides with other substances have great application in the repair of bone tissues [112,113,114]. For example, Sathain et al. mixed CRG, alginate, and calcium silicate together and prepared a biological scaffold that could effectively replace bone tissue defects through freeze-drying technology. The scaffold had good biological activity in vitro and was nontoxic to human cells, and can be used for bone defects repair [109] (Figure 7D). Noroozi et al. prepared a sodium alginate scaffold with suitable pore size and improved mechanical properties, with good application prospects in the repair of bone defects [115].

Due to the excellent characteristics of polysaccharide materials, their application in tissue engineering can be fully exploited. Currently, related polysaccharide materials have been applied in the field of bone tissue engineering, such as chitosan and hyaluronic acid [7,116]. For example, Yang et al. used fiber nanocrystals and alginate as raw materials to prepare a microcapsule using all-water-phase microcapsule electrospray technology; it had good cytoactive and biocompatibility, greatly helped with cell proliferation, and has wide application potential in bone defect repair (Figure 8A). At the same time, they also conducted in vitro tests and found that microcapsules at different concentrations had different promoting effects on DNA, allowing them to gauge the optimal amount for promoting bone defect repair [117] (Figure 8B).

### 4.3. Cartilage Repair 

Cartilage is composed of chondrocytes and intercellular stroma. Cartilage can be divided into elastic cartilage, hyaline cartilage, and fibrocartilage according to the different intercellular substances. However, cartilage injury or loss due to the presence of certain factors can affect the health of the human body. Although the human body itself can self-repair cartilage, in instances of slow repair and serious damaged it is difficult for the human body to repair cartilage on its own. Therefore, cartilage repair is also an important research direction [120,121]. There are many methods for cartilage repair, among which emerging polysaccharide materials have good biocompatibility, degradability, and low toxicity or even nontoxicity, which can be useful in the repair of cartilage tissues [122,123]. For example, hyaluronic acid [124], chitosan [125], and agaroses [126] all play an important role in this aspect. Composite polysaccharide hydrogels have a better repair effect on cartilage. For example, Chen et al. intersected chondroitin sulfate-tyramine (CS–TA) and carboxymethyl pullulan-tyramine (CMP–TA) to prepare a CMP–TA/CS–TA hydrogel with good cytocompatibility that can be used for cartilage repair. In a mouse model, this hydrogel showed good properties. It also showed good testability in water content and DNA detection and provided a new strategy for cartilage regeneration [119] (Figure 8C).

Polysaccharide scaffolds have good repair ability; therefore, they have been widely used in cartilage [54,124]. For example, Zhou et al. prepared silk–CS scaffolds by combining CS with silk fibroin with excellent mechanical properties using salt-soaking, freeze-drying, and cross-linking methods. The application of the scaffolds significantly reduced the occurrence of chondrocyte inflammation and significantly improved the repair rate of cartilage [119] (Figure 8D). Schütz et al. prepared a biphasic scaffold suitable for the treatment of cartilage loss based on alginate, which provided good environmental conditions for cell organization and could effectively promote the successful regeneration of stem cells in the bone marrow and induce osteogenesis [90] (Figure 6C). At the same time, hydrogels are also beneficial for cartilage repair. Shao et al. used fibroin and chitosan to obtain a biohydrogel, which could effectively improve the accumulation rate of TGF-β. This is useful for the repair of damaged cartilage [29].

### 4.4. Arthritis

Arthritis generally refers to an inflammatory disease that occurs between joints and the surrounding tissues of the body caused by inflammation, infection, and other factors. Common arthritis diseases include osteoarthritis and rheumatoid arthritis, which cause great pain in patients [127]. Therefore, effectively relieving pain or treating arthritis is an area that invited extensive attention from researchers, and polysaccharide materials can play a role in the treatment of arthritis after being processed into drugs [128,129]. For example, Yang et al. modified HAMA and MPC to develop a lubricating drug delivery particle for the treatment of arthritis using microfluidic electrojet technology, and it was more effective in the treatment of osteoarthritis under the condition of being loaded with diclofenac sodium (DS) [130]. Lubrication is important for arthritis patients, but at the same time, the problem of arthritis edema cannot be ignored. In order to reduce the problem of arthritis edema, Fan et al. conducted relevant research in this area and constructed a novel HA/Cur nanomicelle drug against RA. This drug can not only more effectively reduce the edema of arthritis but also better reduce the friction between cartilage surfaces, thus protecting the normalization of the morphology of cartilage; at the same time, it can effectively relieve pain in patients [131] (Figure 9A).

In the treatment of arthritis, HA holds an important position. If a hydrogel is used as a carrier of hyaluronic acid and other therapeutic drugs and then delivered to the site of inflammation, due to the hydrogel properties of the support, the drug can be released in a timely, effective, and slow process. For example, Kim et al. prepared HA–Try injectable hydrogels using HA and tyramine as materials. After activation by EDC and sulfo-NHS, the anti-inflammatory drug, DMT, for RA treatment encapsulated inside the hydrogel could be continuously and slowly released for a month. This indicates that the hydrogel can be used as an effective carrier for the drug treatment of RA [132] (Figure 9B).

### 4.5. Cardiovascular System

The cardiovascular system is one of the most important systems in the human body, and deaths caused by cardiovascular diseases continue to increase year by year. Therefore, there is still a long way to go to treat all kinds of cardiovascular diseases. Common cardiovascular diseases include myocarditis, myocardial infarction, stroke, coronary heart disease, and rheumatic heart disease, which seriously affect people’s quality of life. At present, there are many treatment methods for several cardiovascular diseases, among which vascular prosthesis transplantation and artificial blood vessel construction and transplantation is an effective method, but this method is prone to inducing poor functioning or even failure of the artificial blood vessel [135,136]. After modification, heparin and other polysaccharide materials can be widely used in the treatment of cardiovascular system damage and diseases [53,137,138]. For example, the use of heparin-binding protein (HBP) in the treatment of some cardiovascular diseases was outlined by Cai et al. [139]. Shin et al. prepared two kinds of catechol-modified HA (HA-CA) hydrogels and pyrogallol-modified HA (HA-PG) hydrogels with excellent adhesion and biocompatibility by using HA as the material modified by catechol. These two hydrogels can be used to accelerate the regeneration of blood vessels in ischemic areas and have a positive effect on protecting the heart. They can be used as adhesives after partial organ rupture. Moreover, they are also effective in drug delivery and artificial tissue construction [133] (Figure 9C).

The framework of biological scaffolds can be used in the cardiovascular system. For example, Ye et al. prepared nonwoven scaffolds by coupling heparin with PCL by performing electrostatic spinning. The scaffolds had excellent hydrophilicity and compatibility. Compared with PCL alone, the scaffolds had a better loading effect on VEGF and had a positive effect on cell growth and development. It has good application potential in cardiovascular tissue engineering [140]. Wang et al. used 3D printing technology to prepare a black phosphorus (BP) composite fiber scaffold, which can effectively improve vascularization and accelerate the healing of bone defects, and has a broad application prospect in the field of cardiovascular tissue engineering [141].

### 4.6. Nervous System

Among the nine systems of the human body, the earliest developed system is the nervous system, which occurs in the fetal period. Moreover, it is also one of the most important systems and plays a role in controlling and regulating the physiological functions of the body. The system is composed of the brain, spinal cord, cranial nerves, and spinal nerves. Thus, the safety and stability of the nervous system is very important, but neurological damage can still occur. Treating damage to the nervous system has long been a research aim. After modification, polysaccharide materials can play a certain role in the repair of a damaged nervous system [142,143]. For example, Matsumoto’s research group prepared a chitosan nanofiber network management (C-tube) system and carried out experiments in six dogs. First, they removed the right thoracic sympathetic nerve and phrenic nerve of the dogs and then sutured them with the C-tube. After 12 months of the experiment, the corresponding nerves were finally recovered. It was confirmed that the C-tube can be safely and effectively applied in the recovery and regeneration of the damaged phrenic and sympathetic nerves [144]. Tanaka et al. conducted similar experiments following Matsumoto’s process and reached the same conclusion [145]. Yu et al. prepared MC with a chitosan amine group as material, and then formed scaffolds by free radical polymerization and cross-linking. MC scaffolds can enhance neuronal viscosity and promote neurite renewal, which has a good effect in the field of neurology [24].

### 4.7. Reproductive System

Reproduction provides the continuation of human generations. However, an increasing number of people experience the phenomenon that the reproductive system cannot reproduce normally, such as male or female infertility, thin endometrium and other serious phenomena that prevent a person from reproducing normally. The construction and transplantation of artificial ovaries is a new research direction [146,147,148]. At the same time, polysaccharide materials can also effectively help those who cannot reproduce normally [149,150]. For example, Desail et al. prepared an ECM-HA hydrogel by combining tyramine-modified hyaluronic acid with ECM, which can significantly improve the survival rate of follicles in vitro and has good application in follicles requiring in vitro culture [151]. For the occurrence of a thin endometrium, although not as severe an issue as infertility but a noteworthy problem, polysaccharides in the treatment of a thin uterus represent good biological materials. For example, Lie et al. modified VEGF with HAMA under microfluidic electrojet technology and prepared a hydrogel microsphere with good biocompatibility and high controllability that could effectively treat thin endometrium. While improving the drug-loading ability, it could also effectively control the release rate of drugs. It was found that the hydrogel microspheres could effectively promote the normalization of thin endometrium and the restoration of blood vessels [134] (Figure 9D).

Contraception is as important as the repair of the reproductive system. Common methods of contraception include obstacle contraception, hormonal contraception, and surgical contraception, which have their own advantages and disadvantages and are suitable for different situations [152]. The characteristics of chitosan lead to its wide application, and it is also used frequently in reproductive contraception [153]. For example, Luo et al. prepared Chitosan-pcDNA3.1-mCRISp1 polysaccharide nanoparticles. Experimental tests showed that the chitosan nanoparticles not only effectively protected DNA from DNAse I degradation but also greatly reduced the pregnancy rate in mice injected with polysaccharide particles. With the premise of ensuring safety, the contraceptive effect reached the best level among the four groups of experiments, and this polysaccharide nanoparticle has good application prospects in the field of biological reproductive contraception [154].

## 5. Conclusions

Overall, emerging polysaccharide materials have great potential as tissue engineering materials. Polysaccharide materials have certain biocompatibility, biodegradability, stability, ease of processing and modification, and good development and application prospects in tissue engineering and other fields. At the same time, it is also important to combine hydrogel technology with stimuli-responsive polysaccharides to develop scaffolds that can be applied in the field of tissue engineering [155]. However, polysaccharide materials still have certain limitations. The toxicity of the materials prepared by combining polysaccharides with other materials remains to be investigated, and some of the polysaccharide materials have poor mechanical properties and are easily decomposed by foreign substances. Hydrogels made from polysaccharides are still unable to eliminate the poor physical stability caused by hydrogels. Therefore, improving the mechanical properties and biocompatibility of polysaccharide materials represents an important issue. Some polysaccharides have low solubility, and some are susceptible to moisture. The preservation of these polysaccharides remains to be investigated. The use of scaffolds, hydrogels, and microspheres processed from related polysaccharides still needs to be verified in relevant clinical trials.

## 6. Future Perspectives

We hope that researchers will continue to explore the potential of related polysaccharides such as chitosan, cellulose and gellan gum, and prepare them into composite materials according to their unique properties for better application in tissue engineering. Meanwhile, it is hoped that the mechanical properties of some polysaccharides can be improved. Therefore, more practical polysaccharide materials should be developed for tissue engineering research in the future. We hope that this review promotes interdisciplinary communication between tissue engineering and biomaterials and provides an important basis for the research of polysaccharide materials and related fields.

## Figures and Tables

**Figure 1 polymers-14-03268-f001:**
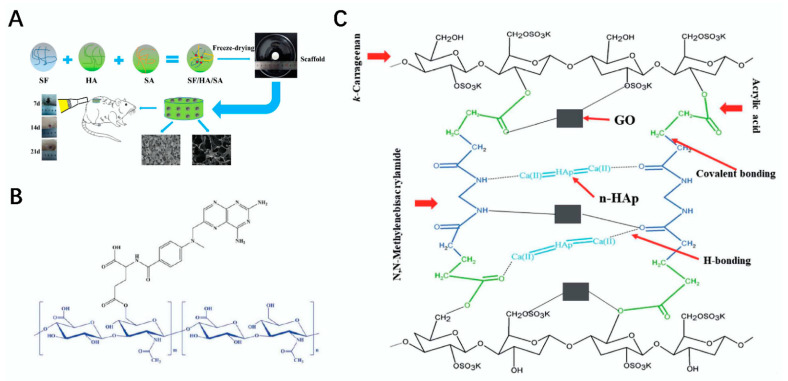
Construction and application of polysaccharide materials and composite scaffolders. (**A**) schematic diagram of the preparation of SF/HA/SA skin scaffolds and their application as wound dressings in a rat full-thickness burn model [21]. (**B**) schematic diagram of the HA–MTX composite [22]. (**C**) schematic diagram of the preparation of the carrageenan, graphene, acrylic acid, and hydroxyapatite composite scaffold [23].

**Figure 2 polymers-14-03268-f002:**
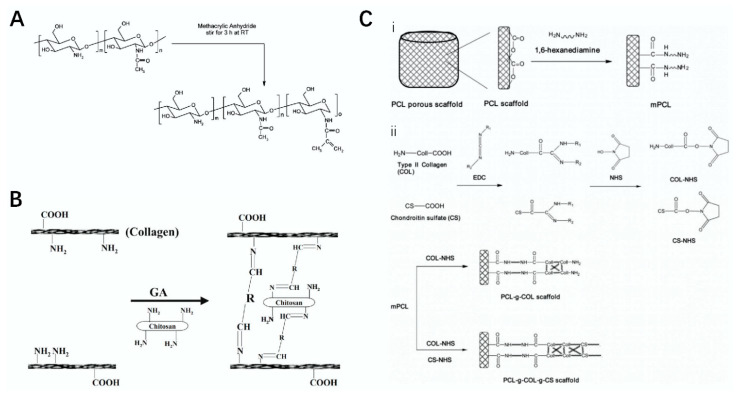
Synthesis roadmap of polysaccharide materials. (**A**) schematic diagram of the preparation of methyl acrylamide chitosan (MC) [30]. (**B**) schematic diagram of the preparation of a glutaraldehyde-modified collagen–chitosan composite scaffold [31]. (**C**) schematic diagram of the preparation of Poly (ε-caprolactone) (PCL)-modified chondroitin sulfate composite scaffold [33].

**Figure 3 polymers-14-03268-f003:**
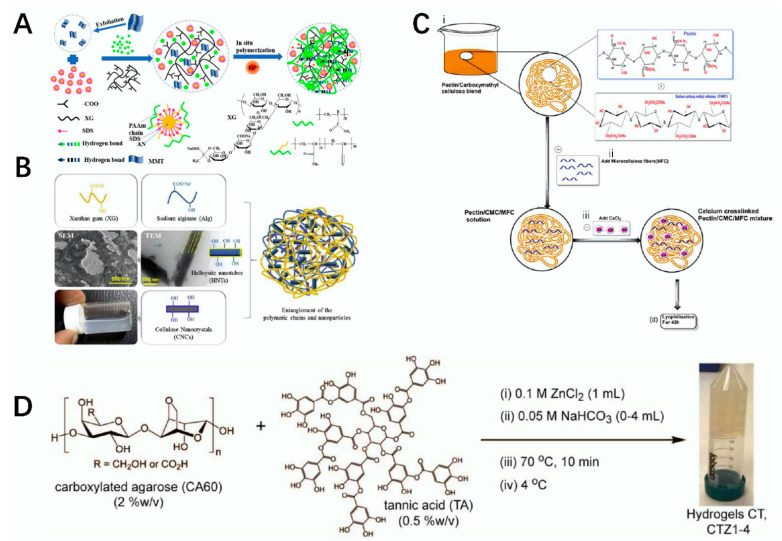
Schematic diagram of the synthesis of polysaccharide materials. (**A**) schematic diagram of the preparation of the XG/MMT/PAAm hydrogel modified by Fe^3+^ [45]. (**B**) schematic diagram of the preparation of the AlgX–CNCs–HNT composite scaffold [44]. (**C**) schematic diagram of the preparation of pectin/CMC/MFC scaffolds [46]. (**D**) schematic diagram of the preparation of CTZ1-4 [47].

**Figure 4 polymers-14-03268-f004:**
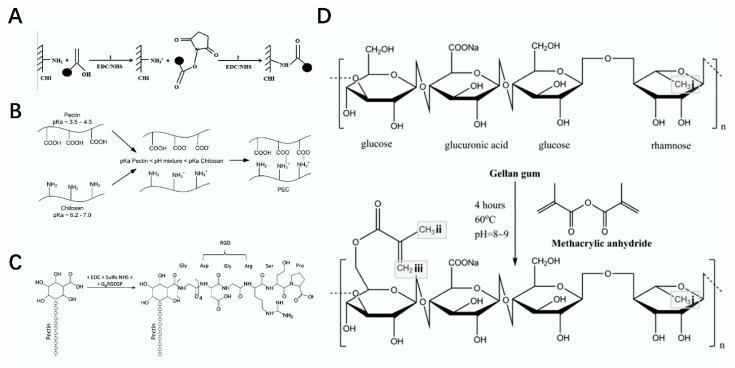
Synthesis roadmap of polysaccharide composites. (**A**) schematic diagram of the preparation of chitosan scaffold modified by heparin [60]. (**B**) schematic diagram of the preparation of the PEC composite scaffold [61]. (**C**) schematic diagram of the preparation of pectin materials [17]. (**D**) schematic diagram of hydrogel modified by methacrylate [62].

**Figure 5 polymers-14-03268-f005:**
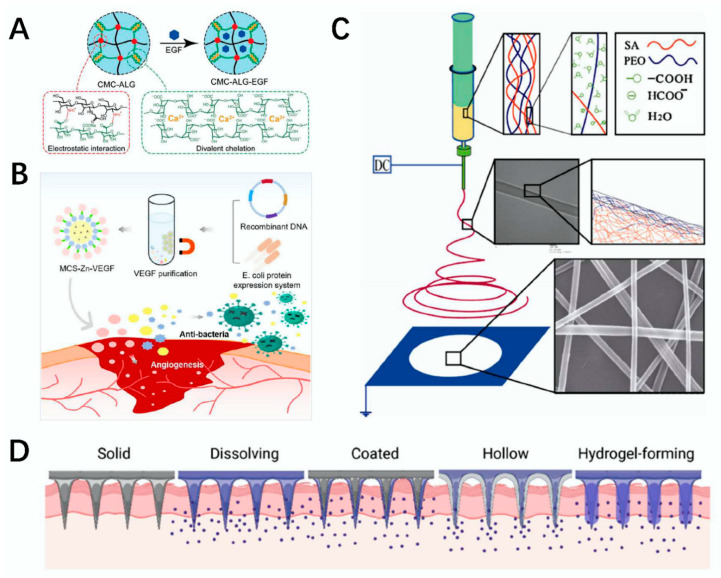
Preparation and application of polysaccharide composite materials and classification of microneedles. (**A**) schematic diagram of an amorphous hydrogel structure made by CMC, SA, and EGF [79]. (**B**) preparation and application of MCS microspheres in wound healing [81]. (**C**) schematic diagram of the preparation of SA/PEO nanofiber membranes [4]. (**D**) classification of microneedles [82].

**Figure 6 polymers-14-03268-f006:**
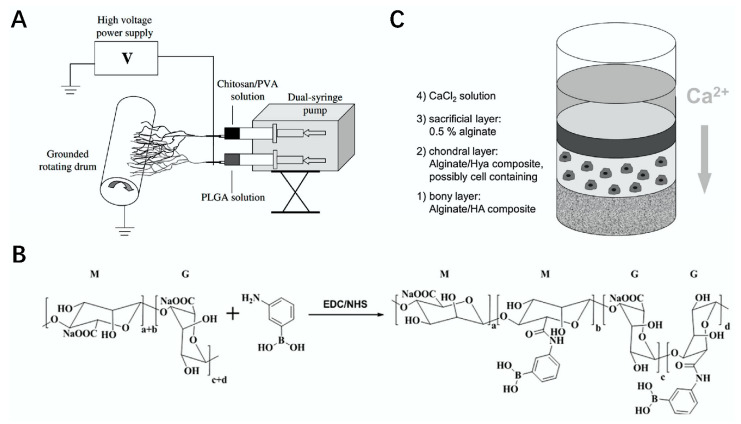
Preparation of materials using related polysaccharides and their applications. (**A**) schematic diagram of the electrospinning of PLGA–chitosan/PVA nanofiber composite film [88]. (**B**) schematic diagram of the preparation of the Alg–APBA composite microneedle [89]. (**C**) bipolar scaffolds prepared with sodium alginate [90].

**Figure 7 polymers-14-03268-f007:**
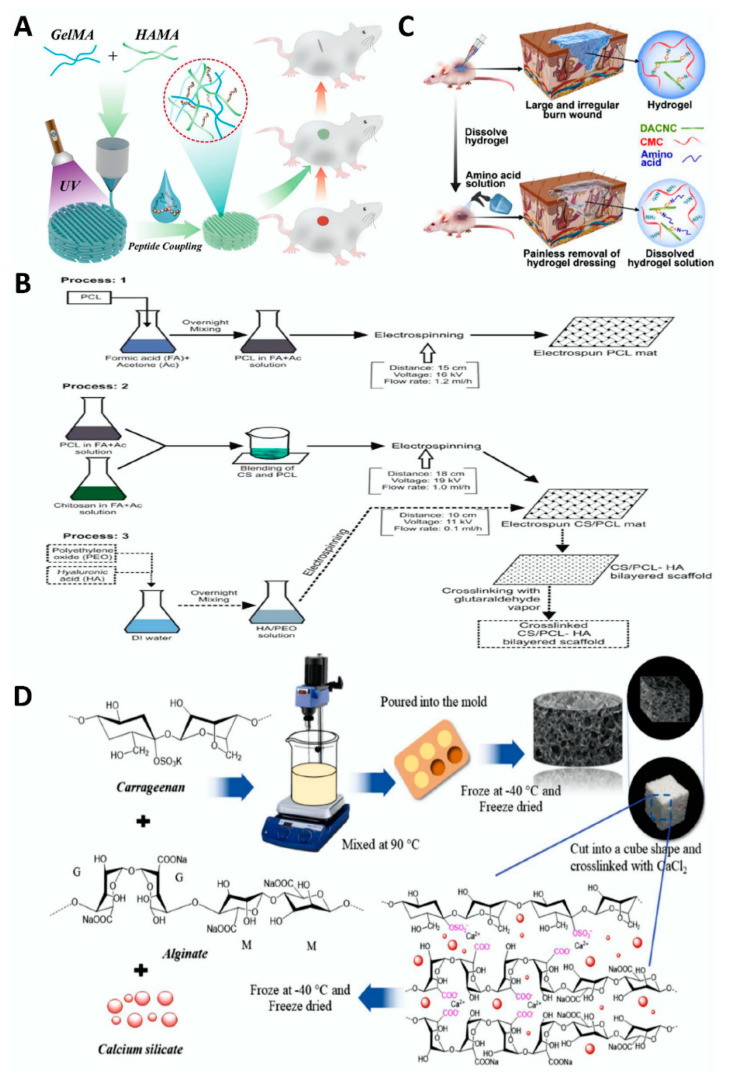
Schematic diagram of the synthesis and tissue engineering application of polysaccharide processed materials. (**A**) schematic diagram of the preparation process of a GelMA/HAMA peptide-coupled patch [106]. (**B**) schematic diagram of the preparation of a CS-PCL-HA double scaffold [107]. (**C**) synthesis and application diagram of hydrogels prepared from DACNC and CMC as materials [108]. (**D**) schematic diagram of preparation of alginate, carrageenan, and calcium silicate composite scaffolds [109].

**Figure 8 polymers-14-03268-f008:**
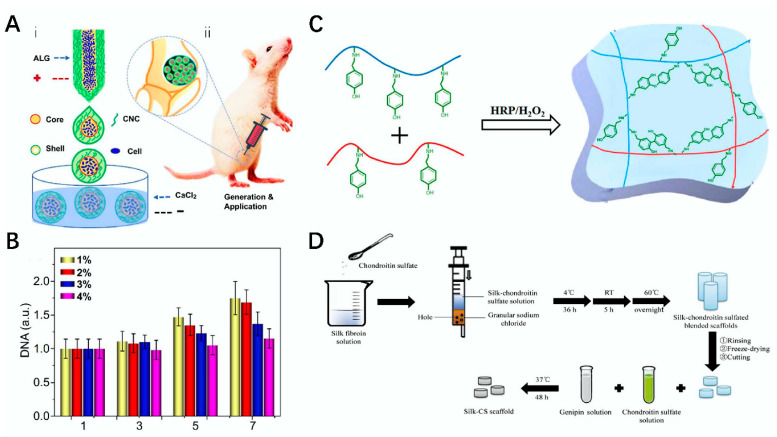
Route map and application of polysaccharide processed into composite materials. (**A**) preparation of core–shell microcapsules and its application in rats [117]. (**B**) effects of microcapsules at different concentrations on cell proliferation [117]. (**C**) schematic diagram of CA-TA and CMP-TA cross-linking gel formation [118]. (**D**) schematic diagram of silk-CS scaffold preparation [119].

**Figure 9 polymers-14-03268-f009:**
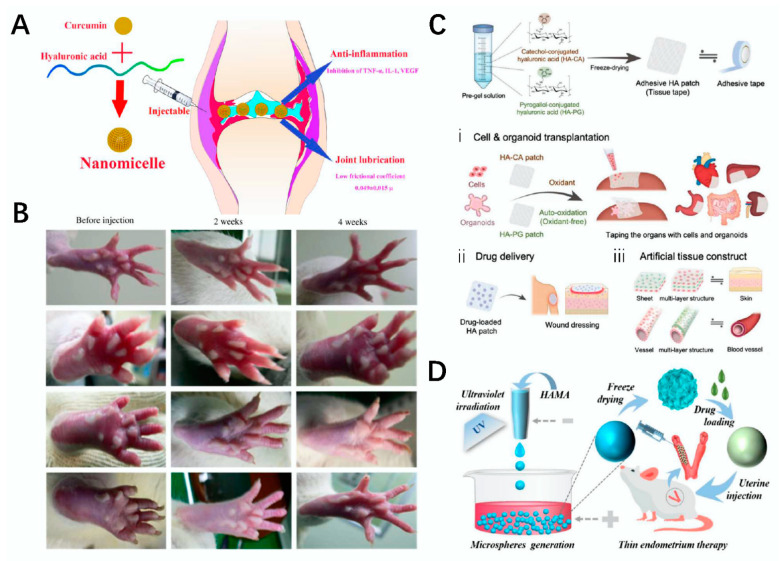
Construction of related polysaccharide materials and its tissue engineering application. (**A**) schematic diagram of the construction of the HA/Cur compound drug and its application in arthritis repair [131]. (**B**) schematic diagram of the HA-Try hydrogel applied in rats [132]. (**C**) schematic diagram of the synthesis and application of HA-CA and HA-CP [133]. (**D**) schematic diagram of the preparation of hydrogel microspheres for the treatment of thin uterus [134].

**Table 1 polymers-14-03268-t001:** Structural formula of polysaccharides.

Name of Polysaccharide	Structural Formula	References
Sodium alginate	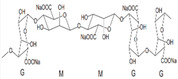	[12]
Chitosan	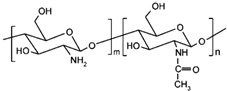	[3]
Hyaluronic acid	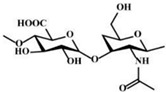	[3]
6-chondroitin sulfate	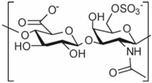	[13]
κ-carrageenan	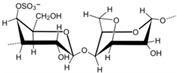	[5]
Xanthan gum	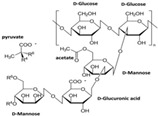	[14]
Cellulose	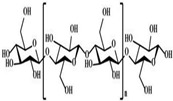	[3]
Agarose	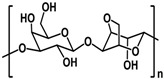	[15]
Heparin	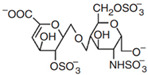	[16]
Pectin	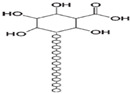	[17]
Gellan gum	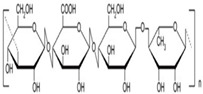	[18]

**Table 2 polymers-14-03268-t002:** Types, schematic diagram and characteristics of polysaccharide derived materials.

Types	Amorphous Hydrogel	Microspheres	Membrane	Fiber	Microneedles
Schematic diagram	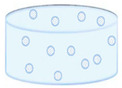	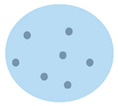	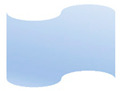	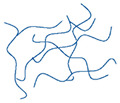	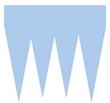
Security	Safety	Higher security	Safety	Low toxic	Safety
Processing difficulty	Easy	Easy	Easy	Easy	Easy
Biocompatibility	Good	Good	Good	Good	Aponia
Trauma	Small	Small	Small	Invasive small	Traumatic small
Other characteristics	Injectable	Low toxic	Selectivity	Workability	Invasive

## Data Availability

The data presented in this study are available in the article.

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
