# Peer review of "Research Progress on Emerging Polysaccharide Materials Applied in Tissue Engineering"

_polymers, 2022, doi:10.3390/polym14163268_

Round 1
Reviewer 1 Report
Respected Authors,
It is a pleasure to accept the task of reviewing your Manuscript ID polymers-1821873, entitled “Research progress on emerging polysaccharide materials applied in tissue engineering applications”. In this paper, recent progress in emerging polysaccharide materials is outlined, by considering tissue engineering applications. This review will promote interdisciplinary communication between tissue engineering and biomaterials and will provide an important basis for the research of polysaccharide materials.
The overall quality of the review is quite good, however, there following shortcomings that should be rectified before acceptance.
1. Most all portions of review articles contain outdated references and is missing some latest references related to polysaccharide materials and tissue engineering.
2. In line 46-47, the authors wrote “There are many more examples of the application of emerging polysaccharide materials such as this in tissue engineering, as mentioned in the following article.” What is meaning of following article? No reference is present there. The authors should clarify this.
3. The introduction is perfectly written; however, some latest studies are missing. The authors are advised to incorporate these studies in their manuscript which related to natural polymeric sustainable materials.
https://doi.org/10.1016/j.fpsl.2022.100892
4. Polysaccharide materials and derivatives section, Line 63-77 contains a large amount of information with only cited reference. These lines must be supported by some reference.
5. Authors should incorporate the novelty of using polysaccharide materials for tissue engineering application. For this a few paragraphs related to degradation needed to be included.
6. In line 88-89, the authors wrote “Niranjan et al. prepared a PVP/SA/TiO2-CUR composite patch based on SA and PVP using a gel-casting method”. Full form of PVP is missing. Abbreviations should be defined at the first mentioned. There are numerous other instances that require perusal of the authors.
7. In 2.7 cellulose section, references are cited seldomly in this subsection and it is missing an important study related to cellulosic biomaterials
8. Applications section looks intriguing and comprehensive with in-vivo literature. Well done for your efforts. However, polysaccharide composites materials are developed by incorporating these materials with synthetic polymers such as PLA, authors are advised to take the following studies as reference and write about use of different polymeric composites in bone, cartilage, nerve, and cardiovascular tissue engineering applications. Try to add some numerical simulation data as well.
https://doi.org/10.1088/1748-605X/ac7308
9. In future prospect, authors should incorporate the latest technology used to develop tissue engineering scaffolds by using stimuli-responsive polysaccharide materials.
10. Similarly, the combination of natural/green fibers and natural polymers has currently gained significant attention in biomedical applications owing to biodegradable and biocompatible nature. This manuscript significantly lacks that particular touch. Authors are advised to see the below references for this purpose:
https://doi.org/10.1016/j.rineng.2021.100263
11. Overall quality throughout the manuscript must be improved by drawing some flow diagrams with the help of modern software.
12. Minor corrections
Please check and revise for typos mistakes.
There are a lot of grammatical mistakes which require perusal of the authors.
Mention full form of abbreviations at first mention.
Once authors agree to take my points into account, I will be happy to accept the manuscript.
In the last, I would like to say Polymers is a very competitive journal, which published high-quality review articles related to biomedical engineering. Based on the aforementioned comments, my recommendation is Major Revision.
Reviewer 2 Report
/
Manuscript 1821873 has a good plan, is offering an appropriate selection of figures and references but the presentation, in the actual form, isn’t permitting the publication.
The English language is poor for most than 25% of the document (very long or incomplete phrases, repetition, pleonasm, inappropriate use of some terms – i.e. lines 123-124, 165, 222, 313, 329, 361, 397/398, 428, 509, 516, 477-479, figures caption, the ) making the text unclear, confuse, even unintelligible sometimes. The repetition of the sentence “The structure of …. is shown in Table 1” can be avoided in an elegant mode. The formula of chitosan in Table 1 must be better replaced by the structure from Fig 2A (specify why). There are many abbreviations not explained (i.e. MMT, VEGF, HAMA, GelMA etc). HA is used both for hyaluronic acid and hydroxiapatite. The abstract must be reformulated. The last phrase is recommended as aim in Introduction or in Conclusions, not there.
Important information missing (i.e. specific characteristics and biological properties of some polysaccharides recommending them for a specific application, tissue engineering requirements and general technologies/techniques enumeration for scaffolds development, importance of such materials in 3D bioprinting etc.) it is difficult for the reader to understand the role of such materials in tissue engineering domain. The discussion quality is unequal, because the authors are not pointing on the role of polysaccharides in the final effect. Advantages and disadvantages are presented briefly in Conclusions, and not developed for every type of polysaccharide to help researchers to make a choice. Writing mistakes are frequent. The authors must check also the references (32, 40, 45, 47, 67, 69).
However, being a comprehensive work it must be reviewed, carefully checked (for language, used terms in respect to some definitions and chemistry), modified to respond to the publishing requirements.
Reviewer 3 Report
In this article, authors described the research progress on polysaccharides used in tissue engineering application. I urge the authors to incorporate these suggestions:
1. There are some grammatical mistakes in manuscript. Please read the manuscript thoroughly.
2. Cite the following related articles
https://doi.org/10.1002/pat.4688, doi: 10.1039/c9ra05025f
3. Clearly state the novelty of the review as there are similar published data in literature.
4. Write the full word where you first use the abbreviation. Don't use the full word again in text.
3. Cite some latest articles of respective journal.
Reviewer 4 Report
The article by Su et al. deals with polysaccharide materials applied in tissue engineering applications. The topic is interesting and timely for the journal. However, there are previous review articles on the field. See for instance: Polysaccharide-Based Biomaterials in Tissue Engineering: A Review. Tissue Eng Part B Rev 2021, 27, 604. Therfore, there is lack of noveltuy in the current article, hence I cannot recomend it for publication in the journal.
Other issues to be addressed can be found below:
The abstract is not concise. Should be summarized. There are too many repetitions. Also, it should be written in thrid person
The English grammar is not accurate. Please ask a native speaker to revise the whole text.
SOme sentences need revision. See for instance in line 34 " polysaccharide materials made from polysaccharide"
line 46 "such as this in tissue engineering"
Many sentences need references. See for instance:
line 34-36
line 37-40
line 48-50
lnie 70-75
Format is not correct. For instance, at the begining of each section/subsection please use capital letter.
table 2 is ood. There should not be figures. Please make a diagram instead.
concluisons and propospects should be separated. Prospects should be called "future perspectives" and this section should be expanded.
some references should be added like: J Mater chem B 2016, 4, 600-612; ACS Applied Materials Interfaces 2016, 20, 17902-14.
Round 2
Reviewer 1 Report
The review article is in the acceptable form now.
Reviewer 4 Report
I do really appreciate the efforts of the authors to improve the work in accordance to my comments. Therefore, although I still believe the novelty is not too high, I will recommend it for publication in the journal.